# The Potential Contribution of the IL-37/IL-18/IL-18BP/IL-18R Axis in the Pathogenesis of Sjögren’s Syndrome

**DOI:** 10.3390/ijms26104877

**Published:** 2025-05-19

**Authors:** Dorian Parisis, Julie Sarrand, Muhammad Soyfoo

**Affiliations:** Department of Rheumatology, Hôpital Erasme, Hôpital Universitaire de Bruxelles HUB, Université Libre de Bruxelles ULB, 1070 Brussels, Belgium; dorian.parisis@ulb.be (D.P.); julie.sarrand@ulb.be (J.S.)

**Keywords:** interleukin-37, interleukin-18, primary Sjögren’s syndrome, salivary glands, autoimmune disease, immunohistochemistry, inflammation biomarkers

## Abstract

The objective of this study was to explore the expression profile of the Interleukin (IL)-37/IL-18/IL-18BP/IL-18R axis in patients with primary Sjögren’s syndrome (pSS). This study included 36 patients diagnosed with pSS, 13 patients presenting with sicca symptoms without confirmed pSS, and 14 healthy controls. Serum concentrations of IL-37, IL-18, IL-18BP, and IL-18R were measured using a sandwich ELISA. These levels were then correlated with relevant clinical and biological parameters. Furthermore, expression of the same cytokines was assessed in salivary gland biopsies via immunohistochemistry. No significant difference in serum IL-37 levels was observed among the three groups (*p* = 0.1695). However, serum levels of IL-18 and IL-18BP were significantly elevated in pSS patients compared to healthy controls (*p* < 0.0001), and these levels were strongly correlated. Immunohistochemical analysis revealed significantly higher expression of IL-37 in both the excretory ducts and inflammatory infiltrates of salivary glands in pSS patients compared to sicca patients. No correlation was found between IL-37 expression and the histological severity of glandular infiltration as assessed by the Chisholm score. In addition, an enhanced expression of IL-18, IL-18BP, and IL-18Rα was observed in the salivary glands of pSS patients. These findings suggest the potential contribution of the IL-37/IL-18/IL-18BP/IL-18R signaling axis in the pathogenesis of Sjögren’s syndrome, particularly through its increased expression in salivary glands and correlation with disease-specific inflammatory markers. These findings may contribute to a better understanding of pSS immunopathology and suggest new avenues for biomarker development or therapeutic targeting.

## 1. Introduction

Sjögren’s syndrome (SS) is a chronic autoimmune disease characterized by lymphocytic infiltration of the salivary and lacrimal glands, resulting in the classical sicca symptoms of xerostomia and keratoconjunctivitis sicca. These symptoms manifest as a range of discomforts: dry eyes often lead to burning, foreign body sensation, and blurred vision due to decreased tear production and tear film instability, while dry mouth can cause difficulties in speaking, chewing, and swallowing, along with an increased risk of dental caries and oral infections. SS may occur as a primary disorder or as secondary, in association with other autoimmune diseases such as rheumatoid arthritis or systemic lupus erythematosus. In addition to the hallmark features of dry eyes and dry mouth, SS is frequently associated with systemic manifestations including arthralgia, fatigue, and multi-organ involvement [1,2].

While there has been significant progress in elucidating the pathophysiological mechanisms underlying SS, it is currently hypothesized that the convergence of genetic predisposition, environmental triggers, and hormonal imbalances may initiate the activation of the normally quiescent epithelium. This activation may lead to the upregulation of toll-like receptors, secretion of damage-associated molecular pattern molecules (DAMPs), increased cell death and apoptosis, and the subsequent release of pro-inflammatory cytokines that perpetuate autoimmune inflammation [3,4,5].

Interleukin (IL)-37 is a member of the IL-1 cytokine family, initially discovered through in silico methods in 2000, though its biological functions were elucidated 10 years later [6,7]. In contrast to other members of the IL-1 family, IL-37 serves as a potent suppressor of both innate and adaptive immune responses, notably by dampening the production of pro-inflammatory cytokines and promoting tolerogenic properties in dendritic cells [6,8]. Transgenic mice overexpressing IL-37 exhibit enhanced resistance to various inflammatory conditions such as septic shock [6], colitis [9,10], ischemia–reperfusion injury [11,12], and contact dermatitis [8].

There are five known splice variants of IL-37 (IL-37a–e), with IL-37b being the most extensively studied, as it contains five of the six exons [13]. The pro-form of IL-37 is activated via cleavage by caspase-1 [14], an enzyme that is also found to be overexpressed in the early stages of Sjögren’s syndrome and whose levels correlate with increased apoptosis in epithelial cells [15].

IL-37 exhibits a dual mechanism of action, functioning both extracellularly as a cytokine and intracellularly as a transcription factor [13]. Extracellularly, IL-37 interacts primarily with the IL-18 system—comprising IL-18, Interleukin-18 receptor (IL-18R), and Interleukin-18 Binding Protein (IL-18BP)—a pathway typically involved in the induction of interferon (IFN)-γ in synergy with IL-12. Despite structural homology with IL-18, IL-37 binds IL-18Rα without recruiting the co-receptor IL-18Rβ and thus fails to induce IFN-γ production. The IL-37/IL-18Rα complex may exert anti-inflammatory effects by preventing the recruitment of the orphan receptor single immunoglobulin IL-1R-related receptor (SIGIRR), as shown by Nold-Petry et al. [16]. Although IL-37 has a low affinity for IL-18Rα and cannot function as a competitive inhibitor of IL-18 [13], it can bind to IL-18BP—a natural soluble inhibitor of IL-18—enhancing its suppressive effect by impeding the stabilization of the IL-18/IL-18Rα complex by IL-18Rβ [13].

Once cleaved by caspase-1, IL-37 translocates to the nucleus, where it acts as a transcription factor by interacting with Mothers Against Decapentaplegic Homolog 3 (SMAD3) in the Transforming Growth Factor (TGF)-β signaling pathway, thereby exerting immunosuppressive effects [8]. IL-37 establishes a negative feedback loop with a range of pro-inflammatory cytokines, including IL-18, IFN-γ, TNF, IL-17, IL-1α, IL-1β, and IL-6. It also inhibits dendritic cell activation and promotes the expansion of regulatory T cells, thereby mitigating excessive inflammatory responses [8].

The role of IL-37 has been investigated in several autoimmune diseases, including Guillain–Barré syndrome [17], Crohn’s disease [18], Graves’ disease [19], psoriasis [20], rheumatoid arthritis [21], systemic lupus erythematosus [22], and ankylosing spondylitis [23]. However, to date, the expression and role of IL-37 have not been studied in the context of Sjögren’s syndrome. This study aims to explore the contribution of the IL-37/IL-18/IL-18BP/IL-18R signaling axis to the immunopathogenesis of primary Sjögren’s syndrome, with a focus on its local and systemic expression profiles.

## 2. Results

### 2.1. Clinical and Demographic Characteristics of Patients

A total of 36 pSS patients, 13 sicca patients, and 14 healthy controls were included in this study. There were no significant differences in age or sex ratio among the groups. The average age of the pSS group was 57.2 ± 2.3 years, with a female-to-male sex ratio of 8:1, which does not differ significantly from the 9:1 ratio reported in the literature [1].

Clinically, there were no significant differences in the incidence of arthralgia, myalgia, Raynaud’s phenomenon, or paresthesia between the pSS and sicca groups.

Biologically, a significantly higher presence of rheumatoid factor was observed in the pSS group compared to the sicca group (*p* = 0.0468), indirectly reflecting B lymphocytic dyscrasia.

Subgroup analysis revealed a statistically significant association between the presence of autoantibodies and ESSDAI score (*p* = 0.0467).

The characteristics of the study groups are summarized in Table 1.

### 2.2. Increased Serum Concentrations of IL-18 and IL-18BPa, but Not IL-37, in pSS Patients

Serum concentrations of IL-37 did not differ significantly between the pSS, sicca, and healthy control groups (*p* = 0.1695). Serum IL-18 levels were significantly elevated in both the pSS and sicca groups compared to the healthy controls (*p* < 0.0001), but no significant difference was observed between the pSS and sicca groups (*p* = 0.5794). Serum IL-18BPa levels were significantly elevated in both pSS and sicca patients compared to healthy controls (*p* < 0.0001 and *p* = 0.0108, respectively), and were also significantly higher in pSS patients than in those with sicca symptoms (*p* = 0.0111) (see Figure 1).

### 2.3. Correlation Study Between Serum Levels of IL-18, IL-37, and IL-18BPa and Demographic, Clinical, and Laboratory Parameters

We examined whether the serum levels of these cytokines correlated with age, focus score, or ESSDAI, and whether these parameters correlated among themselves.

Serum IL-18 and IL-18BPa levels were positively correlated (r = 0.456, *p* = 0.0002) (Figure 2A).

No correlation was observed between IL-37 and IL-18 or IL-18BPa levels.

Serum IL-18BPa levels were also positively correlated with age (r = 0.484, *p* < 0.0001) (Figure 2B).

Subgroup analysis of the pSS cohort revealed no differences in relation to Chisholm score, sex, the presence or absence of anti-SSa/SSb autoantibodies, hypergammaglobulinemia, ESSDAI score (≤5 or >5), Schirmer test abnormalities, salivary scintigraphy, arthralgia, or myalgia.

### 2.4. Tissue Expression Profiles of IL-37, IL-18, IL-18Rα, and IL-18BPa in Salivary Glands of pSS Patients

The expression of IL-37, IL-18, IL-18Rα, and IL-18BPa was evaluated by immunohistochemistry in salivary gland biopsies obtained from patients with pSS (n = 12) and from sicca patients (n = 6).

IL-37 expression was higher in pSS patients compared to those with sicca (Figure 3A). This overexpression was observed at the level of the excretory ducts and in inflammatory cells. No difference in expression was noted based on Chisholm score. IL-37 expression was absent at the acinar level.

IL-18 expression was also elevated in pSS patients compared to sicca (Figure 3B), with increased staining in the excretory ducts and inflammatory cells. No correlation with Chisholm score was observed. IL-18 expression was not detected in acinar cells.

IL-18BPa expression at the excretory ducts was significantly higher in pSS patients with Chisholm scores II and III compared to those with Chisholm IV and sicca patients. The highest expression was seen in pSS patients with Chisholm III (Figure 3C). No IL-18BPa expression was observed in acini or infiltrates.

Lastly, IL-18Rα expression at the excretory ducts was significantly higher in pSS patients than in sicca patients (Figure 3D). No difference was observed in relation to Chisholm score. IL-18Rα expression was also present in inflammatory infiltrates.

## 3. Discussion

In this study, we demonstrated that the serum and glandular expression of IL-18 and IL-18BP are elevated in pSS patients compared to healthy controls. Although there were no differences in serum IL-37, we demonstrated via immunohistochemistry that the local expression of IL-37 is increased at the excretory duct level of the salivary glands of pSS patients as compared to sicca patients.

Serum levels of IL-18 were not significantly different between pSS and sicca, even though they were significantly higher than those in healthy controls. The sicca patients included in this study were individuals without infiltration of the salivary glands, without SSa/SSb autoantibodies, and without known infectious, inflammatory, or neoplastic diseases. The term “sicca” encompasses all causes of non-SS dry eye syndrome, making it a heterogeneous group and therefore difficult to study. In our study, their clinical manifestations were not statistically different from those of pSS patients. There may be a selection bias among sicca patients, as rheumatology and internal medicine services predominantly recruit those presenting with “systemic manifestations”. These elevated levels of IL-18 and IL-18BPa compared to healthy controls may reflect undiagnosed inflammatory disease or even a “pre-Sjögren” state.

In this study, we report that serum levels of IL-18 and IL-18BPa are significantly elevated in patients with pSS compared to those with sicca symptoms, with a positive correlation observed between the two cytokines. IL-18 has been the subject of several studies in pSS [24,25,26,27,28]. Eriksson et al. [24] and Bombardieri et al. [25] reported significantly increased serum IL-18 levels measured by ELISA in pSS patients compared to healthy controls, associated with myalgia in the former and the presence of autoantibodies in the latter. We did not demonstrate such associations in our cohort. Chen et al. [26] reported, in 2015, a significantly increased expression of IL-18 in pSS patients (especially with ESSDAI scores > 7) compared to controls, but no difference in serum IL-18BP. Our results differ, but it should be noted that the average age in our pSS group is higher (57.19 ± 13.66 vs. 44.43 ± 12.88), and we observed a correlation between IL-18BP and age, which may explain this difference.

We report that the expression of IL-18 is increased in the salivary glands of pSS patients compared to sicca patients. In pSS, this expression is predominantly located at the level of the excretory ducts and in inflammatory infiltrates, with no difference in expression based on Chisholm score. The upregulation of IL-18 expression in the salivary glands of pSS patients has already been demonstrated by immunohistochemistry [25,27,28,29]. According to these studies, IL-18 is expressed at the excretory ducts of the salivary glands and in CD68+ macrophages within periductal foci. The expression of IL-18 in inflammatory infiltrates is particularly notable when these foci are organized into tertiary lymphoid structures [24], and is positively correlated the focus score, the number of B and T lymphocytes, and glandular swelling, and inversely correlated with serum complement levels [30]. Conversely, no IL-18 expression was observed in infiltrates from chronic sialadenitis [24]. This suggests that IL-18 may play a role in the early stages of the disease, similar to other alarmins such as IL-33 [31,32].

This study is the first to evaluate the glandular expression of IL-18Rα and IL-18BPa in the salivary glands of pSS patients using immunohistochemistry. We report here that IL-18BPa expression is higher in patients with Chisholm scores II and III. We did not observe the protein expression of IL-18BPa in pSS patients with Chisholm IV. A similar expression pattern based on Chisholm score has previously been reported for IL-33 in pSS [31].

Our study is also the first to investigate the role of IL-37, an anti-inflammatory cytokine of the IL-18 subfamily, in pSS. IL-37 has already been studied in other rheumatic diseases such as rheumatoid arthritis (RA) [6,21,33], lupus [34,35], and ankylosing spondylitis [22,36]. All of these studies converge toward the same conclusion: serum IL-37 levels significantly increase in widespread systemic inflammatory autoimmune diseases. Our study did not demonstrate an elevation of IL-37 in the serum of patients with systemic forms of pSS (ESSDAI > 5), nor a correlation between IL-37 and IL-18 levels. However, we demonstrated by immunohistochemistry that the local expression of IL-37 is increased at the excretory duct level of the salivary glands in pSS patients compared to sicca patients. Given the retrospective nature of our study, it should be noted that patients with systemic forms of pSS had previously been treated with corticosteroids and immunosuppressive therapies at the time of serum collection, which may explain the absence of elevated serum IL-37 levels.

The IL-18/IL-37/IL-18BPa/IL-18Rα axis, via the induction of IFN-γ, appears to play a central role in the pathogenesis of Sjögren’s syndrome. The expression of IL-37 and IL-18 by excretory ducts, even in the absence of lymphocytic infiltrate, may reflect the activation of these structures. The secretion of IL-18, insufficiently counterbalanced by IL-37 and IL-18BP, leads to IFN-γ secretion by numerous cells of the innate and adaptive immune systems, such as antigen-presenting cells, cluster differentiation (CD)4+ and CD8+ T lymphocytes, B-cells, and natural killer (NK) cells. IFN-γ is directly implicated in the overexpression of Human Leukocyte Antigen (HLA)-DR, B-cell Activating Factor (BAFF), and other co-stimulatory factors at the excretory ducts, as well as in promoting apoptosis. This increased apoptosis is associated with elevated caspases (including 1 and 8), First apoptosis signal receptor (Fas)/Fas ligand (FASL) signaling (including sensitivity via CD40), and the downregulation of the anti-apoptotic factor c-FLIP under the influence of IFN-γ. The apoptosis of epithelial cells leads to the formation of apoptotic blebs rich in SSa, SSb, and Sm antigens, which are then taken up by antigen-presenting cells, initiating an adaptive immune response (HLA-DR dependent) against these autoantigens. The resulting production of autoantibodies and memory cells leads to the formation of immune complexes, which subsequently interact with toll-like receptor (TLR)9 and FcR-γ, contributing to the persistence and amplification of the autoimmune response.

Finally, these findings should be interpreted considering some limitations. First, the tissue analysis cohort was relatively small (12 pSS and 6 sicca patients), which may limit the generalizability of our histological findings. Future studies with larger sample sizes will be necessary to validate these results and further explore cytokine expression patterns in salivary glands. Second, the sicca group is inherently heterogeneous, comprising individuals with varying degrees of dryness symptoms but without a confirmed diagnosis of pSS. This heterogeneity may introduce confounding variables and complicate direct comparisons with the pSS group. A more detailed phenotypic and immunologic characterization of the sicca group, along with longitudinal follow-up, would help assess the risk of progression to overt autoimmune disease and clarify their place within the disease spectrum.

## 4. Patients and Methods

A total of 36 patients with pSS, defined according to the criteria of the American–European Consensus Group in 2002 [37], 13 patients with dry eye syndrome without autoimmune involvement (sicca), and 14 healthy controls were included in this study. Patients with other inflammatory, infectious, or neoplastic conditions were excluded. Local inflammatory involvement of the salivary glands was assessed using Chisholm scores, and systemic disease activity was evaluated using **E**uropean **S**jögren’s **S**yndrome **D**isease **A**ctivity **I**ndex (ESSDAI) scores. Clinical data were collected, including medical history, current medications, and various laboratory parameters. The study was approved by the local ethics committee (Comité d’Ethique Hospitalo-Facultaire Erasme-ULB), and informed consent was obtained from all patients and control subjects.

### 4.1. Cytokine Determination by ELISA (Enzyme-Linked Immunosorbent Assay)

Serum expression of IL-37 (dilution 1:2) was measured using an enzyme-linked immunosorbent assay (ELISA) kit provided by Adipogen AG (Liestal, Switzerland), following the manufacturer’s protocol. Serum levels of IL-18 (dilution 1:5) and IL-18BPa (dilution 1:10) were measured using ELISA kits commercially available from R&D Systems (Abingdon, UK), also according to the manufacturer’s instructions.

### 4.2. Immunohistochemistry

The expression of IL-37, IL-18, IL-18Rα, and IL-18BPa was analyzed by immunohistochemistry in salivary gland biopsies from pSS patients (n = 12) and sicca patients (n = 6). Tissue sections were deparaffinized through successive baths of toluene and isopropanol. Endogenous peroxidase activity was blocked with a 0.3% hydrogen peroxide solution in methanol for 30 min. After rinsing with distilled water, slides were heated in citrate buffer (pH 6) in a microwave (800 W for 10 min) for antigen retrieval, then blocked with 10% Normal Goat Serum in TBS for 1 h. The sections were then incubated overnight at 4 °C with the following primary antibodies: anti-IL-37 (mouse monoclonal, Abcam; 1:300, Cambridge, UK), anti-IL-18 (mouse monoclonal, R&D Systems; 1:1500), anti-IL-18BPa (mouse monoclonal, R&D Systems; 1:100), and anti-IL-18Rα (mouse monoclonal, R&D Systems; 1:1500), diluted in 1% Normal Goat Serum in Tris-Buffered Saline (TBS). After washing with TBS 1 × (3 × 10 min), sections were incubated with biotinylated goat anti-mouse secondary antibodies (dilution 1:200, Vector Labs, Newark, CA, USA) for 1 h. Following another wash with TBS, the sections were treated with streptavidin–peroxidase (ABC kit, Vector Labs). Detection was carried out using DAB (diaminobenzidine), followed by a final TBS wash. The sections were then counterstained with Mayer’s hematoxylin, dehydrated, and mounted using 1,3-diethyl-8-phenylxanthine (DPX).

### 4.3. Statistical Analysis

All statistical analyses were performed using GraphPad Prism 6.4 software. The normality of the data distribution was assessed using the Shapiro–Wilk test. For normally and non-normally distributed data, Student’s *t*-test and the Mann–Whitney test were applied, respectively. Spearman’s correlation coefficient was used to assess correlations between clinical variables, biological variables, and cytokine concentrations. A *p*-value < 0.05 was considered statistically significant.

## 5. Conclusions

Our findings support a potential role for the IL-37/IL-18/IL-18BP/IL-18R signaling axis in the pathogenesis of pSS, notably through its enhanced expression in salivary gland tissue and its association with disease-specific inflammatory markers. These insights may advance our understanding of pSS immunopathology and open new avenues for the development of diagnostic biomarkers and targeted therapeutic strategies.

## Figures and Tables

**Figure 1 ijms-26-04877-f001:**
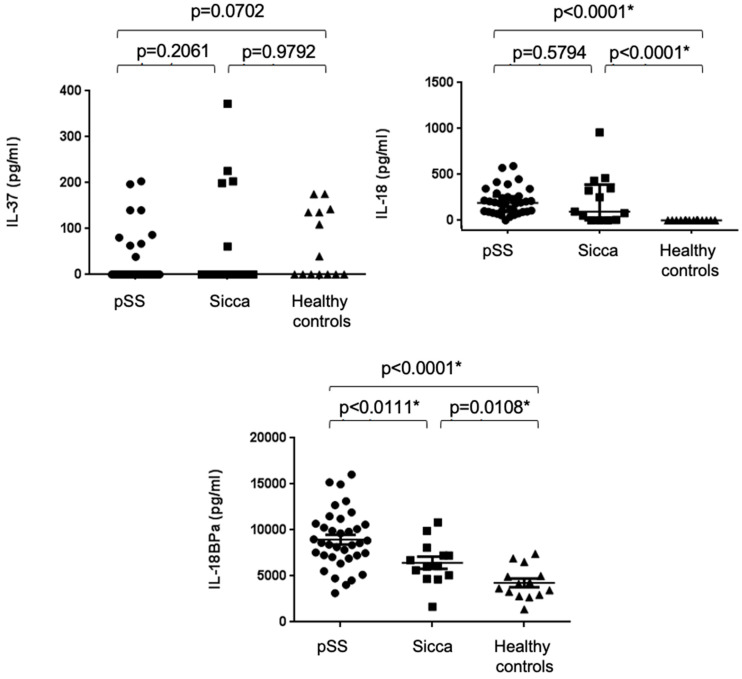
Serum levels of IL-18, IL-18BPa, and IL-37 in pSS, sicca, and healthy controls. IL-18 and IL-18BPa significantly elevated in pSS and sicca compared to controls. IL-18BPa also higher in pSS than in sicca. IL-37 levels showed no significant differences. The asterisk represents statistical significance.

**Figure 2 ijms-26-04877-f002:**
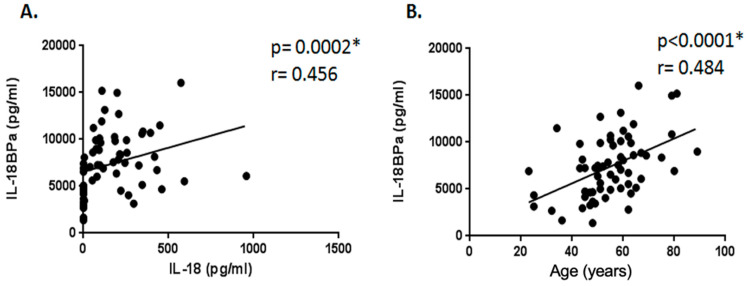
Correlation between serum cytokine levels and clinical parameters. (**A**) Positive correlation between serum IL-18 and IL-18BPa levels (r = 0.456, *p* = 0.0002). (**B**) Positive correlation between IL-18BPa levels and age (r = 0.484, *p* < 0.0001). No significant correlation observed between IL-37 and either IL-18 or IL-18BPa. The asterisk represents statistical significance.

**Figure 3 ijms-26-04877-f003:**
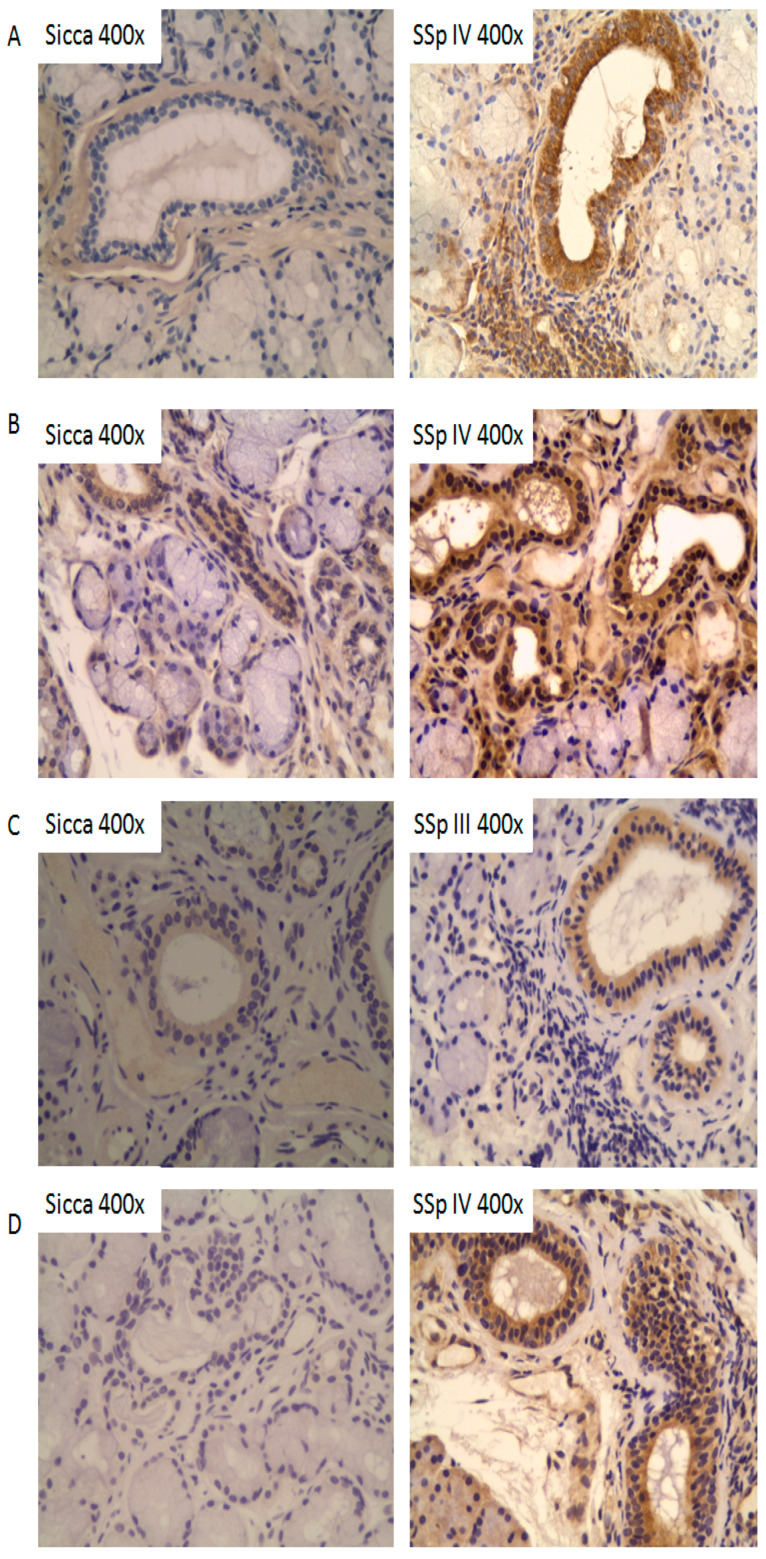
The immunohistochemical analysis of IL-37, IL-18, IL-18BPa, and IL-18Rα expression in salivary gland biopsies from pSS and sicca patients. (**A**) IL-37 expression was increased in pSS patients, localized to excretory ducts and inflammatory cells. (**B**) IL-18 expression was elevated in pSS patients, predominantly in excretory ducts and infiltrates. (**C**) IL-18BPa expression was significantly increased in pSS patients with Chisholm scores II and III, especially at the excretory duct level. (**D**) IL-18Rα expression was also higher in pSS patients, observed in both excretory ducts and inflammatory infiltrates. No staining was observed in acini, and expression levels did not vary significantly with Chisholm score, except for IL-18BPa.

**Table 1 ijms-26-04877-t001:** Clinical, demographic, and biologic characteristics of patients and controls.

		SS (N = 36)	Sicca (N = 13)	Healthy Controls (N = 14)	*p*
**Demographic data**					
Age (years)	Mean ± SEM	57.2 ± 2.3	54.2 ± 3.2	49.50 ± 3.416	0.1765
Sex ratio	n (F:M)	32 (8:1)	12 (12:1)	13 (13:1)	0.8834
**Diagnosis criteria**					
Chisholm ≥ 3	n (%)	24/33 (72.7%)	0		
Focus score	Mean ± SEM	1.4 ± 0.2	0		
SSA and/or SSB	n (%)	19/34 (55.9%)	0		
Ocular signs	n (%)	25/32 (78.1%)	6/8 (75%)		1
Oral signs	n (%)	12/22 (54.5%)	3/7 (42.9%)		1
Ocular symptoms	n (%)	36 (100%)	12 (92.3%)		0.2653
Oral symptoms	n (%)	34 (94.4%)	11 (84.6%)		0.2839
**ESSDAI**	Med [range]	1.424 [0–21]			
**Clinical**					
Raynaud	n (%)	4 (11.1%)	2 (15.4%)		0.6495
Arthralgia	n (%)	25 (69.4%)	7 (53.8%)		0.3309
Myalgia	n (%)	10 (27.8%)	5 (38.5%)		0.5001
Paresthesia	n (%)	8 (22.2%)	3 (23.1%)		1
**Biologic**					
ANA +	n (%)	26/35 (74.3%)	0 (0%)		<0.0001 *
Inflammatory syndrome	n (%)	9/35 (25.7%)	1/12 (8.3%)		0.2479
RF	n (%)	9/34 (26.5%)	0 (0%)		0.0468 *
Hyperγ	n (%)	8/34 (23.5%)	0 (0%)		0.0855
Cryoglobulinemia	n (%)	3 (8.3%)	0 (0%)		0.5555
**IL-37/IL-18/IL-18BP axis**					
IL-37 (pg/mL)	Med [range]	0 [0–203.2]	0 [0–372.1]	0 [0–175.6]	0.1695
IL-18 (pg/mL)	Med [range]	190.3 [0–593.2]	96.09 [0–957.3]	0 [0–5]	<0.0001 *
IL-18 BP (pg/mL)	Mean ± SEM	8958 ± 517.3	6447 ± 657.5	4251 ± 467.7	<0.0001 *

Abbreviations: **ANA**: Antinuclear Antibodies; **BP**: Binding Protein; **ESSDAI**: EULAR Sjögren’s Syndrome Disease Activity Index; **F**: Female; **Hyperγ**: hypergammaglobulinemia; **IL**: Interleukin; **M**: Male; **med**: Median; **pg/mL**: picograms per milliliter; **RF**: rheumatoid factor; **SEM**: Standard Error of the Mean; **SSA/SSB**: Anti-Sjögren’s Syndrome-related Antigen A/B; **SS**: Sjögren’s syndrome. The asterisk represents statistical significance.

## Data Availability

Data is contained within the article.

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
