# Peer review of "The Potential Contribution of the IL-37/IL-18/IL-18BP/IL-18R Axis in the Pathogenesis of Sjögren’s Syndrome"

_ijms, 2025, doi:10.3390/ijms26104877_

Round 1
Reviewer 1 Report
Comments and Suggestions for Authors
This is an interesting study that analyses IL-37, IL-18, IL-18BP and IL-18R protein expression in the serum and salivary glands of Sjogren, and sicca patients as well as healthy controls. The references used are appropriate. The article is mostly well written and clear; however, some comments and suggestions are provided below:
- Cytokine levels in the serum are known to vary because of circadian rhythm. How did the authors take this factor into account in their study?
- The authors state that the study was approved by the local ethics committee. More details could be provided i.e. what was the name of the ethics committee?
- In table 1 the numbers should be written with a decimal point e.g., 57.2 not 57,2.
- It would be helpful if the table had a footnote to explain abbreviations used in the table.
- The writing is quite clear however, there appears to be French words in the article e.g. et/ou, sains. The article should be checked to ensure all words are written in English.
- Ocular signs and oral signs mentioned in table 1 could be explained.
- Are some p values missing from table 1 (e.g. SSA, SSB)?
- Use of mean or median could be made clearer.
- Overall, the information in table 1 could be made clearer.
- Line 100, p=0.11695 whereas in table 1 p=0.1695. Is there an error?
- Why does the y-axis in figure 1 (IL-18) not start at 0?
- Subheadings could be made clearer e.g., lines 111-112.
- Line 150 Figure 3b should be written as Figure 3B
- In the introduction IL-37 is described as being an anti-inflammatory cytokine yet in the discussion it is reported that levels are elevated in various inflammatory autoimmune disorders. Suggest that the authors discuss this aspect more.
The English is mostly of high quality however there are French words in the article. This should be corrected.
Author Response
This is an interesting study that analyses IL-37, IL-18, IL-18BP and IL-18R protein expression in the serum and salivary glands of Sjogren, and sicca patients as well as healthy controls. The references used are appropriate. The article is mostly well written and clear; however, some comments and suggestions are provided below:
Cytokine levels in the serum are known to vary because of circadian rhythm. How did the authors take this factor into account in their study?
We appreciate the reviewer’s insightful comment regarding the potential influence of circadian rhythms on serum cytokine levels. The timing of sample collection was generally restricted to morning hours ( between 8:00 AM and 10:00 AM), which we believe helped to reduce, circadian-related variability. Importantly, because this time frame was consistent across study groups, any residual circadian influence is unlikely to have introduced systematic bias that would differentially affect the comparisons central to our conclusions. We recognize this as a limitation and have now included it in the discussion of the manuscript.
The authors state that the study was approved by the local ethics committee. More details could be provided i.e. what was the name of the ethics committee?
The name of the ethics committee has been added.
In table 1 the numbers should be written with a decimal point e.g., 57.2 not 57,2.
It has been corrected
It would be helpful if the table had a footnote to explain abbreviations used in the table.
It has been added
The writing is quite clear however, there appears to be French words in the article e.g. et/ou, sains. The article should be checked to ensure all words are written in English.
It has been corrected
Ocular signs and oral signs mentioned in table 1 could be explained.
It has been added in intro
Are some p values missing from table 1 (e.g. SSA, SSB)?
No p-values were reported for SSA and SSB in Table 1 because these variables were compared to a reference value of zero, making statistical testing not applicable in this context. To clarify this, we have added a dash (–) in the table to indicate that no p-value is provided by design.
Use of mean or median could be made clearer.
Overall, the information in table 1 could be made clearer.
The presentation of Table 1 has been improved by standardizing abbreviations, clearly specifying whether values represent means or medians, and using a dash (–) to indicate when no data are provided by design.
Line 100, p=0.11695 whereas in table 1 p=0.1695. Is there an error?
Yes there is an error, the correct value is 0.1695
Why does the y-axis in figure 1 (IL-18) not start at 0?
It was a typing mistake, we changed it to 0
Subheadings could be made clearer e.g., lines 111-112.
Subheadings have been added to the results
Line 150 Figure 3b should be written as Figure 3B
It has been changed
In the introduction IL-37 is described as being an anti-inflammatory cytokine yet in the discussion it is reported that levels are elevated in various inflammatory autoimmune disorders. Suggest that the authors discuss this aspect more.
We thank the reviewer for this important remark. In the discussion, we do acknowledge that IL-37 levels are elevated in several autoimmune diseases. However, we also clearly state that this increase appears insufficient to counterbalance the heightened pro-inflammatory environment. Specifically, we explain that the secretion of IL-18, inadequately neutralized by IL-37 and IL-18BP, leads to increased IFN-γ production by multiple immune cell types (e.g., antigen-presenting cells, CD4⁺ and CD8⁺ T cells, B cells, and NK cells). We believe this explanation addresses the apparent paradox of IL-37 elevation in autoimmune conditions.
The English is mostly of high quality however there are French words in the article. This should be corrected.
It has been corrected.
Reviewer 2 Report
Comments and Suggestions for Authors
This manuscript represents a valuable and timely study investigating the role of the IL-37/IL-18/IL-18BP/IL-18R signaling axis in the pathogenesis of primary Sjögren’s syndrome. The integration of systemic and local analyses, particularly the novel assessment of IL-37 expression in salivary glands, makes an important contribution to this topic. The methodology is robust, the statistical analyses are performed appropriately, and the results are well contextualized in the existing literature.
The study provides important insights, but minor revisions are recommended to improve clarity and strengthen the manuscript:
- Table and Figure Improvements:
Table 1: For clarity, please add a list of abbreviations used in the table.
Figure 2: Review the labeling of the x-axis to ensure accuracy and consistency.
- Sample size:
The tissue analysis cohort (12 pSS and 6 sicca patients) is relatively small. Future studies with larger samples would be beneficial to confirm these results.
- Heterogeneity of the sicca group:
A more detailed characterization of the sicca group is recommended to clarify possible confounding factors and assess the risk of progression to pSS.
Author Response
This manuscript represents a valuable and timely study investigating the role of the IL-37/IL-18/IL-18BP/IL-18R signaling axis in the pathogenesis of primary Sjögren’s syndrome. The integration of systemic and local analyses, particularly the novel assessment of IL-37 expression in salivary glands, makes an important contribution to this topic. The methodology is robust, the statistical analyses are performed appropriately, and the results are well contextualized in the existing literature.
The study provides important insights, but minor revisions are recommended to improve clarity and strengthen the manuscript:
Table and Figure Improvements:
Table 1: For clarity, please add a list of abbreviations used in the table.
It has been done
Figure 2: Review the labeling of the x-axis to ensure accuracy and consistency.
It has been done
Sample size:
The tissue analysis cohort (12 pSS and 6 sicca patients) is relatively small. Future studies with larger samples would be beneficial to confirm these results.
Heterogeneity of the sicca group:
A more detailed characterization of the sicca group is recommended to clarify possible confounding factors and assess the risk of progression to pSS.
These 2 aspects have been added as limitation of the study in discussion